# Design and Massaging Force Analysis of Wearable Flexible Single Point Massager Imitating Traditional Chinese Medicine

**DOI:** 10.3390/mi13030370

**Published:** 2022-02-26

**Authors:** Zhou Zhou, Yixuan Wang, Chenjun Zhang, Ao Meng, Bingshan Hu, Hongliu Yu

**Affiliations:** 1Institute of Rehabilitation Engineering and Technology, University of Shanghai for Science and Technology, Shanghai 200093, China; 1819640932@st.usst.edu.cn (Z.Z.); 1819640911@st.usst.edu.cn (Y.W.); 1819640926@st.usst.edu.cn (C.Z.); 1819640922@st.usst.edu.cn (A.M.); yhl98@hotmail.com (H.Y.); 2Shanghai Engineering Research Center of Assistive Devices, Shanghai 200093, China

**Keywords:** traditional Chinese medicine, uni-acupoint massage, contact force, series elastic actuating

## Abstract

In the theory of traditional Chinese medicine, acupoints refer to special points and areas on the meridian line of the human body. Traditional Chinese medicine believes that the application of unique techniques such as pressing, kneading, rubbing, pushing, and patting to acupoints or massage with the help of specific tools has the effects of promoting blood circulation, dredging meridians, and eliminating fatigue. At present, most automatic massage devices are for large-area massage of the trunk, and few are specifically for acupoint massage of the limbs. First, this paper analyzes the characteristics of traditional Chinese medical acupoint massage and then obtains the design index of an automatic acupoint massage device. After that, based on the principle of a series elastic actuating mechanism, a flexible uni-acupoint massage device and control system, imitating the acupoint massage technique of traditional Chinese medicine, were designed. In order to analyze the massage force of the massage device, the man–machine contact dynamic model of the massage device was established, and the force of the massage device was simulated and analyzed. Finally, an experimental platform was built to verify the massage force and massage process of the massage device. The experimental results show that the massage device designed in this paper meets the indexes of traditional Chinese medical massage, in terms of the massage process and massage force, and verify the rationality of the design.

## 1. Introduction

Based on the guidance of the basic theory of traditional Chinese medicine, acupoint massage is a method of assisting treatment that has a long history in China for the prevention and treatment of chronic or degenerative diseases, by using different massage techniques such as pressing, kneading, rubbing, pushing, and patting [1,2]. It has the effects of promoting blood circulation, dredging meridians, eliminating fatigue, relieving pressure, regulating function, and preventing diseases [3,4]. 

Traditional acupoint massage mostly adopts manual operation, and the work is too heavy for traditional Chinese medicine doctors, while it is also difficult for them to ensure a consistency of massage process and force [5]. With the development of technology, researchers continue to develop various automatic massage devices. For example, the researchers from Waseda University and Asahi University in Japan developed a robot called WAO-1 with a massage scheme that can collect and analyze facial data, so as to control two six-degrees-of-freedom (DOF) mechanical arms and realize facial rehabilitation [6,7]; an Israeli company developed a massage robot called Whee Me that looks like a tracked vehicle and uses a tilt sensor to realize massage through its movement [8]; Hu Lei developed a massage robot including a positioning platform and end-effector, which uses position change to control the force, so as to achieve the purpose of massage [9]; Lu Shouyin developed a traditional Chinese medical massage robot composed of an adjustable massage platform and two five-DOF manipulators [10]; Jenjira Chimsa designed a massage device based on the ischemic compression massage method [11]; the researchers from Chengdu University of information technology developed a wearable massage robot Anmoji, which can massage the head, neck, back, legs, and feet [12]; Huang, Yuancan developed a four-DOF humanoid manipulator that can realize three massage actions: pressing, kneading, and plucking [13].

From the above research status, it can be seen that the existing automatic massage devices are mostly for large massage areas, which makes the overall structure of the massage device large, cumbersome, and complex. However, there is a clear argument for the following: while, patients can move their limbs, and, thus, can very accurately place the limbs relative to large massaging machines, it is difficult to continuously target the massage force on a point in a specific way, which will lead to inaccurate massaging and low massage efficiency. In addition, devices for acupoint massage will have multiple-DOF, resulting in a complex structure and difficulty in miniaturization. However, it is difficult for a large-scale massage devices to massage a single acupoint of the limbs. Considering some massage devices mentioned above and in the literature [14,15], it can be concluded that the current large-scale massage devices are mostly for back massage. Besides, most of the existing massage devices directly realize the massage function through mechanical extrusion or vibration. Moreover, they lack the guidance of traditional Chinese medical massage theory and do not simulate massage techniques such as pressing, kneading, and releasing of traditional Chinese medical massage, so it is difficult to ensure the curative effect of acupoint massage [16]. Furthermore, the current massage devices mostly use a rigid structure [17] when contacting with the human body, which makes it difficult to ensure the essential safety of human–machine interaction in the process of massaging.

To solve the problems existing with the current massage devices, this paper designs a flexible uni-acupoint massage device that can accurately apply massage force to the uni-acupoint of limbs and simulate the acupoint massage process of traditional Chinese medicine. The rest of this paper is organized as follows: the second section analyzes the acupoint massage techniques of traditional Chinese medicine, and obtains the technical indicators and work flow for an acupoint massage device; in the third section, the mechanism and control system of flexible uni-acupoint massage device are designed; in the fourth section, the human–machine contact dynamic model of uni-acupoint massage device is established; in the fifth section, the human–machine contact force is simulated and experimentally analyzed; finally, application of the uni-acupoint massage device is verified.

## 2. Analysis of Acupoint Massage Techniques in Traditional Chinese Medicine

The curative effect of uni-acupoint massage in traditional Chinese medicine is closely related to the parameters of acupoints, direction, strength, action frequency, and action duration [18,19]. After collecting some main data on traditional Chinese medical acupoint massage, it was found that different masseurs have certain rules on the main parameters of acupoint massage. According to the massage techniques described in [20,21], the data shown in Table 1 are summarized, which shows the main parameters of typical pressing methods and kneading methods for limb massage. 

The ‘pressing method’ is to press the patient’s acupoints with fingers or palms and gradually press down with force. The ‘kneading method’ uses the rib surface of the finger, thenar, or palm to perform circular kneading on the acupoints. As can be seen from Table 1, the pressing method and kneading method are movements with 1-DOF. The pressing method moves vertically along the normal direction of the acupoints, and kneading method rotates along the tangential plane of the acupoints. In terms of massage force, the massage force is less than 80 N. There are great differences in the frequency of the massage methods, in which the frequency of finger pressing is 3–10 times/min and the frequency of finger kneading is 60–80 times/min.

In the actual massage process, the masseur will use different massage techniques alternately, to achieve the effect of all-round massage on the human tissue [22]. For example, when massaging a uni-acupoint on the limbs, the commonly used technique of a masseur is a combination of pressing, kneading, and releasing. That is, first press and hold the acupoint with the finger pulp for a period of time. Then, the action of finger kneading is to press and hold the acupoint with the finger pulp and rotate it. Finally, releasing is the action of separating the fingers from the acupoints, so as to relax the acupoints. The purpose of alternating different massage techniques is to prevent the acupoints from becoming insensitive or even vulnerable to damage due to maintaining the same massage method for a long time. Long-term massage can obtain a therapeutic effect. The uni-acupoint massage device designed in this paper simulates the acupoint massage process of the above traditional Chinese medical massage. First, the head of the massage device applies force downward perpendicular to the plane where the acupoint is located, and the force is strictly controlled in the range 10–80 N, from light to heavy, and when reaching the maximum contact force, it is maintained for 10 s, to simulate finger pressing. Then the massage head makes a continuous rotary motion to drive the subcutaneous tissue to produce a corresponding motion, so as to generate friction between the tissue layers. The force of rotation is uniform and strictly controlled in the range of 10–80 N, the rotation speed is 60 rpm, and the total rotation time is 10 s, simulating the action of finger kneading. Finally, it slowly withdraws the force, lets the human tissue relax for 10 s, and repeats the above process. The whole process combines the characteristics of finger pressing and finger kneading, which has a good effect on eliminating blood stasis, accumulation, and regulating blood circulation for local pain acupoints. At the same time, constant-pressure static and no-load relaxation processes are added to the massage process to realize the strong-weak-strong change process of human stimulation, so that the human tissues can fully digest the effect of the process; and so is does not cause damage due to a sudden high load. The technical indicators of the entire process are shown in Table 2.

## 3. Design of a Flexible Uni-Acupoint Massage Device 

### 3.1. Mechanism Design of a Uni-Acupoint Massage Device

According to the above analysis, the typical acupoint massage process mainly includes two actions: pressing and kneading. Therefore, the uni-acupoint massage device designed in this paper also includes two-DOF. Figure 1 shows a 3D model of the flexible uni-acupoint massage device. The body of the massage device is composed of a massage head, a fixed belt, a massage head pressing mechanism, and a massage head rotating mechanism. The pressing mechanism simulates the pressing action of traditional Chinese medical masseur during acupoint massage, and the rotating mechanism simulates the kneading action. The massage device is fixed to the human limbs through a fixed belt. The fixed belt should be a wide and inelastic cloth, to prevent additional loads from affecting the massage effect of the device during fixation.

In order to control the contact force of the massage device, one method is to add an expensive force sensor to the massage device for closed-loop control, but this will undoubtedly increase the cost [23]. The uni-massage device designed in this paper uses the principle of series elastic actuating for referencing [24], and designs the massage head pressing mechanism as shown in Figure 2. The massage head pressing mechanism comprises a motor, a reducer, a worm-and-gear, guide rods, a moving plate, and a spring. When the motor rotates, the reducer and the worm will convert the rotation of the motor into the up and down linear movement of the moving plate, and the massage head is connected to the moving plate through a spring. Therefore, the compression of the spring can be controlled by controlling the stroke of the motor, and finally the contact force between the massage head and the skin is controlled. At the same time, due to the elastic mechanism, the massage device is in flexible contact when contacting with limbs, which makes the massage process more comfortable and safer. In addition, we investigated the rotation speed of the hand of the masseur in the massage process, simulated this in the design, and set the speed to a constant 66 rpm. Since the massage force is effective within a certain range, the impact of acceleration on the actual work effect is temporarily ignored in this study.

Figure 3 shows the rotating mechanism of the massage head. When the pressing mechanism keeps the massage head in contact with the acupoints, a direct-current reduction motor with rated output torque is used to drive the massage head to rotate, so as to provide a stable finger kneading massage force. In addition, this device follows the traditional Chinese medicine finger kneading massage and has a bionic design for the massage head. During finger kneading massage, the point at which the masseur applies force is not fixed, but rotates around the fixed axis, and the massage force applied to the patient using the finger abdomen (to increase the force bearing area) is not perpendicular to the contact surface, but has an inclination angle of nearly 45 degrees with the contact surface. The purpose of this is to fully and efficiently drive the friction between the subcutaneous tissues at the acupoints, so as to obtain the expected therapeutic effect. Therefore, a bias convex massage point is designed to imitate the above finger movement techniques. This can be divided into two parts. One is a semicircle for connecting with the massage head, and the other one is an ellipse with large curvature for imitating the finger abdomen. At the same time, there is a large angle included between the massage point and the middle shaft of the spring within the allowable range of the mechanism.

### 3.2. Design of Control System for the Uni-Acupoint Massage Device

Figure 4 shows the principle diagram of the control system of the acupoint massage device in this paper. The main components include a Microcontroller Unit (MCU), 2 direct-current motor drive chips, power conversion chips, light-emitting diode (LED), switch, and keys. The MCU collects the status of the switch and the keys, and outputs the pulse width modulation (PWM) signal to the motor drive chip, according to the set massage process and massage force, to control the movement of the two motors. In order to be portable and wearable, and to reduce space, this article specially designed two massage device control circuit boards. One of them mainly realizes the human–machine interaction function and integrates the keys, LED, and power interface. The other one integrates a MCU and motor drive chips. A flat cable is used for data transmission between the two circuit boards.

This device uses three key switches to adjust the massage force: key1, key2, and key3 correspond to massage forces of 15 N, 30 N, and 45 N, respectively. The MCU outputs three PWM signals to control the three operating states of the two motors. PWM0 controls the rotating action of the rotating mechanism, PWM1 controls the relaxing action of the pressing mechanism, and PWM2 controls the pressing action of the pressing mechanism. 

As shown in (Figure 5), when key1 is pressed, the acupoint massage device operates according to gear 1. First, the PWM2 output makes the worm gear motor run for 10 s. At this time, the massage head slowly moves to the acupoints until it touches the acupoints and applies massage force. After 10 s running time, 15 N massage force is applied. Then the three-way PWM is fully closed for 10 s, and the massage head contacts the acupoints and applies a constant force to keep the acupoints in a non relaxing low-intensity stimulation state. After that, PWM0 outputs a 100% duty cycle to make the rotating motor run for 10 s, drive the massage head to perform a rotary movement, simulate the action of kneading, and drive the subcutaneous tissue, to produce a corresponding movement. Finally, PWM0 and PWM2 are closed, and PWM1 output makes the pressing motor run in reverse for 10 s, to return the massage head to the initial position. In this process, the massage force gradually decreases until it disappears. When there is no new signal input, the system will repeat the above process. 

The difference between the three keys is that the duty cycle of PWM signal applied to the pressing motor is different. The duty cycles of key2 and key3 are 2 and 3 times that of key1, respectively. In this way, the pressing motor moves at a faster speed, so that within 10 s of pressing, the downward travel of the motor is 2 and 3 times that of key1, respectively, thus, changing the pressing pressure.

## 4. Establishment of a Contact Dynamic Model

In order to analyze the massage force applied by the device to the acupoints during acupoint massage, it is necessary to establish an acupoint massage device–human body coupling dynamic model. Two models of the acupoint massage device and human skin tissue should be considered.

Since the uni-acupoint massage device is based on the series elastic actuating principle, the influence of the damping term can be ignored; a simplified dynamic model of the acupoint massage device is shown in Figure 6. That is, the motor of the pressing mechanism changes the position *S* of the mobile platform *M_1_* through the worm gears, correspondingly drives the compression of the spring *K_1_*, and finally causes the massage head *M_2_* to move, so as to output a force controlled by the compression amount of spring *K_1_* to the massage head.

Human skin is composed of layers (epidermis, dermis, and subcutaneous tissue) with different components and thicknesses. Each layer has a specific mechanical effect. Its model is complex and changeable, and it is difficult to accurately test and analyze the specific pressure generated by the human body when the acupoint massage device is working. However, according to the Kelvin model, human skin mainly shows viscoelasticity when subjected to an external force, so its action model can be simplified and made equivalent to a single-layer viscoelastic model: contact stiffness *K_2_* is used to represent its elastic part, and viscous damping *D* is used to represent its viscous part, as shown in Figure 7 [25].

Equations (1) and (2) can be used to calculate the values of contact stiffness and viscous damping coefficient in the equivalent model of human skin [26]. Taking the forearm of the upper limb as an example, the relevant parameters of the forearm skin were obtained through a dynamic micro imprinting experiment (forearm skin contact stiffness *K*_1_ = 165.02 n/m, viscous damping coefficient *D =* 0.22 ns/m).
(1)$$K=\left|\frac{f_{0}}{u_{0}}\right|\cos\left(\varphi \right){+m\omega }^{2}$$
(2)$$\omega D=\left|\frac{f_{0}}{u_{0}}\right|\sin\left(\varphi \right)$$

When the device acts on human skin, an acupoint massage–human body equivalent dynamic model can be obtained through the mechanical model and the equivalent model of human skin pressure, as shown in Figure 8.

In the series elastic mechanism, the output force *F* is directly proportional to the shape variable *∆X* of the elastic element, that is,
*F* = *k*∆X*(3)

Through the stress analysis of load *M_2_*, it can be obtained
(4)$$M_{2}{g\;+\;K}_{1}x_{1}\;-\;(K_{1}+K_{2})x_{2}\;-\;b\dot{x_{2}}=2\ddot{x_{2}}$$
where *x*_1_ represents the displacement of the moving plate *M*_1_, *x*_2_ represents the displacement of massage head *M*_2_; *K*_1_ and *K*_2_ represent the spring stiffness and the equivalent stiffness coefficient of human skin tissue in the uni-acupoint massage device, respectively; *b* represents the equivalent damping coefficient of human skin tissue; $\dot{x_{2}}$ is the derivative of displacement of mass block *M*_2_ with respect to time; $\ddot{x_{2}}\;$ is the derivative of $\dot{x_{2}}$ with respect to time.

The system shown in Figure 8 was analyzed by using the state space method, and the state vector *z* was defined to correspond to the displacement of mobile platform *M*_1_, the displacement, and speed of load *M*_2_, respectively:(5)$$z=\left[\begin{array}{c} z_{1}\\ \begin{array}{c} z_{2}\\ z_{3} \end{array} \end{array}\right]=\left[\begin{array}{c} x_{1}\\ \begin{array}{c} x_{2}\\ \dot{x_{2}} \end{array} \end{array}\right]$$

Differentiating the state vector *z* with respect to time, the system can be organized into first-order differential equations of each state quantity:(6)$$\left\{\begin{array}{c} \dot{z_{1}}=v\\ \begin{array}{c} \dot{z_{2}}{=z}_{3}\\ \dot{z_{3}}=\frac{1}{M_{2}}{(K}_{1}z_{1}{\;-\;(K}_{1}{\;+\;K}_{2})z_{2}{-\mathrm{bz}}_{3}) \end{array} \end{array}\right.$$

The state space of the uni-acupoint massage device human skin tissue coupling system has the following input equation:$$\overset{\text{˙}}{z}=\mathrm{Az}+\mathrm{Bu}$$
where *A* is the system state matrix, *B* is the input matrix, and *u* is the input vector.
(7)$$A=\left[\begin{array}{ccc} 0 & 0 & 0\\ 0 & 0 & 1\\ \frac{K_{1}}{M_{2}} & -\frac{K_{1}{+K}_{2}}{M_{2}} & -\frac{b}{M_{2}} \end{array}\right]B=\left[\begin{array}{c} 1\\ 0\\ 0 \end{array}\right]u=v$$

For the system model, the displacement of the massage head *M*_2_ is the actual compression amount *∆X* of the spring in the uni-acupoint massage device. Therefore, z_2_ in the state vector is selected as the output of the system.

The output equation of state space is
(8)y=Cz+Du
where *C* is the output state matrix and *D* is the output matrix.
(9)$$C=\left[\begin{array}{ccc} 0 & K_{1} & 0 \end{array}\right]D=0$$

Substitute Formula (9) into Formula (8) to work out the massage force applied to the skin by the uni-acupoint massage device.

## 5. Massage Force Simulation and Experimental Analysis

### 5.1. Friction Experiment

According to the man–machine contact dynamic model in Section 3, a theoretical simulation model is established to simulate and analyze the human–machine contact force. In the simulation, the movement speed of the moving plate in the massage device is 0.85 mm/s, and the stiffness coefficient of the spring in the uni-acupoint massage device is 3.96 N/mm. The stiffness coefficient of human skin tissue is 165.02 N/m, and the damping coefficient is 0.00026 Ns/mm. The mass of the massage head is 0.0217618 kg.

The simulation results are shown in (Figure 9). The abscissa in the Figure is the moving distance of the moving plate driven by the pressing motor, and the ordinate is the pressing force. The black straight line is the pressure when the massage device contacts the rigid contact surface, and the red curve is the pressure simulation result when the massage device contacts the simulated human skin tissue. It can be seen from the Figure, that when the contact surface is a rigid plane, the contact force is directly proportional to the moving distance of the moving plate (i.e., the moving distance of the massage head). When the contact plane simulates human skin tissue, the output load of the massage device shows a certain fluctuation under the influence of viscoelastic damping. When the displacement is within 1 mm, the pressing amplitude is higher than the time acting on the rigid contact surface for a part of the time, due to the influence of contact stiffness, and the amplitude gap gradually increases with the increase of displacement. At 10 mm displacement, the gap reaches 1.88 N, which is 4.75% lower than the contact force of the rigid plane.

To further verify the massage force exerted on acupoints by the uni-acupoint massage device, an experimental platform was set up, as shown in Figure 10. The experimental platform included stents, an uni-acupoint massage device, a silicon rubber plate, and a force sensor. The top surface of the uni-acupoint massage device was fixed on the bracket, and the other side was in contact with silicone rubber; with silicone rubber as a substitute for skin. A six-dimensional force sensor was used to detect the massage force exerted by the uni-acupoint massage device. In the experiment, the moving plate in the uni-acupoint massage device was controlled to move at the same speed of 0.85 mm/s, and the pressing pressure of the six-dimensional force sensor along the pressing direction was recorded, as shown in the blue curve in the figure.

In the experiment, the total weight of the prototype and silica gel pad was about 10 N, which needs to be removed from the recorded force data. It can be seen from the figure that the experimental results match the simulation data of the simulated skin tissue, and the goodness of fit *R^2^* is 0.9809. At the initial stage of starting the massage device, the output force of the prototype experiment was slightly less than that of the human skin simulation; when the moving position of the moving plate from 3 to 7 mm, the experimental results were approximately equivalent; when it was close to 10 mm, the experimental output force of the prototype became gradually less than the simulation results. Compared with the simulation data of the simulated skin tissue, the error was mainly caused by mechanical transmission error and model error between the silicone rubber and real skin tissue. During the whole experiment, the peak output load of the uni-acupoint massage device was 33.71 N. Compared with the results of the rigid plane simulation, the peak difference of output load was 5.89 N, which was reduced by 14.9%.

In the uni-acupoint massage device designed in this paper, the controller adjusts the duty cycle of the control signal PWM2 when the motor is pressed down by reading the value of the keys, so as to achieve the purpose of adjusting the moving distance of the moving plate. The duty cycle of the PWM2 signal corresponding to key2 and key3 is 2 and 3 times that of key1, and control the motor, to drive the moving plate to move at a constant speed that is 2 or 3 times the speed of key1 for a certain period of time. If the acting surface is a rigid contact surface, the moving distance of the moving plate corresponding to key2 and key3 is 2 and 3 times that of key1, and the pressure increases exponentially. However, in contact with human skin tissue, the pressure has non-linear variation. As can be seen from Figure 9, although there is a certain error, the pressure is still nearly 2 times and 3 times. For acupoint massage, the size of the massage force is effective within a certain range, and the control accuracy of the force does not need to be very high. Compared with the cost of other uni-acupoint massage devices, the method of adjusting the massage force selected in this paper, based on the series elastic actuating principle, is desirable.

### 5.2. Wearing Experiment

Wearing experiments were carried out on three healthy male volunteers aged 20, all of whom gave informed consent. As shown in Figure 11, the uni-acupoint massage device was fixed on the forearm with the belt, and the three keys were used to test the human body feeling when the massage forces of different size were run for two minutes (in order to prevent muscle fatigue at the acupoint, the interval of each experiment was 30 min).

From the feedback of volunteers, it was shown that wearing the device feels good and does not excessively affect body movement. With the increase of the selected massage force, the massage feeling obviously changed. When using key1, volunteers said they felt a feeling similar to touching; when using the key2, the volunteers said they had an obvious feeling of pressing and kneading and felt more relaxed in the process; when using key3, volunteers said that they felt a stronger force than when pressing key2, and one of them felt slight pain when the massage time was close to two minutes.

The same experiment was carried out with the uni-acupoint massage device fixed on the leg, and the results were similar to those above. A comparative experiment between healthy and non-healthy people is indeed a very meaningful experiment, but it needs a long observation period, which will be the focus of our subsequent research.

To sum up, the acupoint massage device designed in this paper can basically meet the needs of different people for acupoint massage of the limbs.

## 6. Conclusions

In this paper, according to the action characteristics and mechanical properties of acupoint massage techniques commonly used in traditional Chinese medicine, and aiming at the problem of the single effect of the popular massage devices on the market and the low degree of reduction of professional traditional Chinese medical massage techniques, a flexible uni-acupoint massage device imitating traditional Chinese medical massage techniques was designed. The device is driven based on the principle of series elastic actuating, which can realize two massage techniques: pressing and kneading. In this paper, a human–machine interaction dynamics model of a uni-acupoint massage device was established, and the massage force was simulated and experimentally studied. The simulation and experimental results show that the massage device can well replicate the massage strength and massage process of traditional Chinese medicine masseurs, and can effectively realize uni-acupoint massage of limbs.

## Figures and Tables

**Figure 1 micromachines-13-00370-f001:**
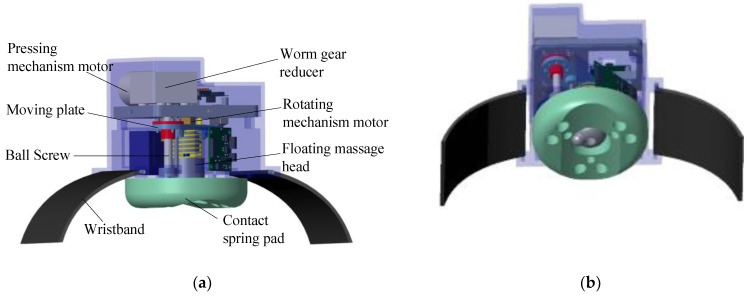
Schematic diagram of the uni-acupoint massage device: (**a**) The overall structure diagram of the flexible uni-acupoint massage device, (**b**) Schematic diagram of massage head.

**Figure 2 micromachines-13-00370-f002:**
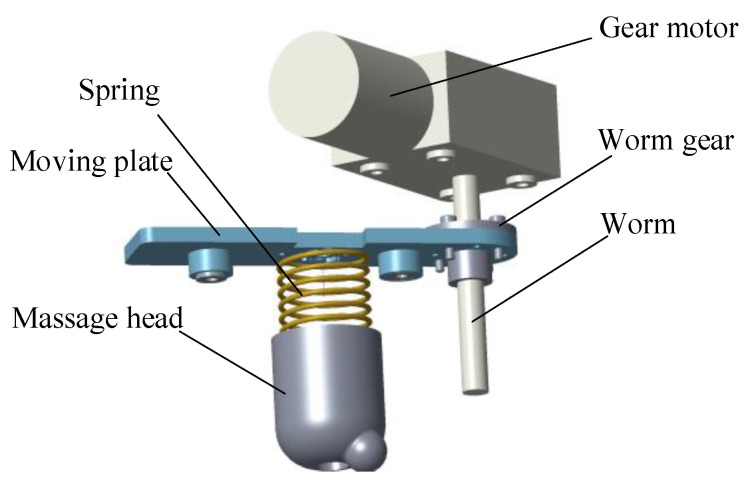
Uni-acupoint massage device massage head pressing mechanism.

**Figure 3 micromachines-13-00370-f003:**
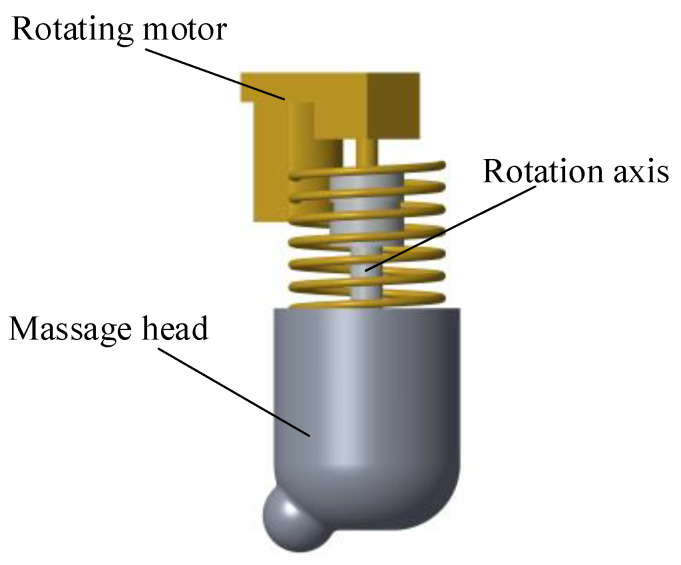
Uni-acupoint massage device massage head rotation mechanism.

**Figure 4 micromachines-13-00370-f004:**
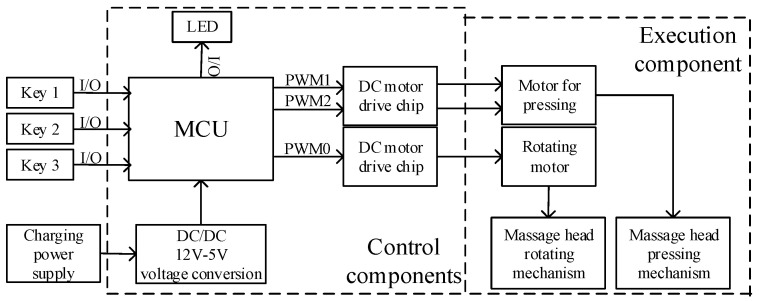
Overall block diagram of control system.

**Figure 5 micromachines-13-00370-f005:**
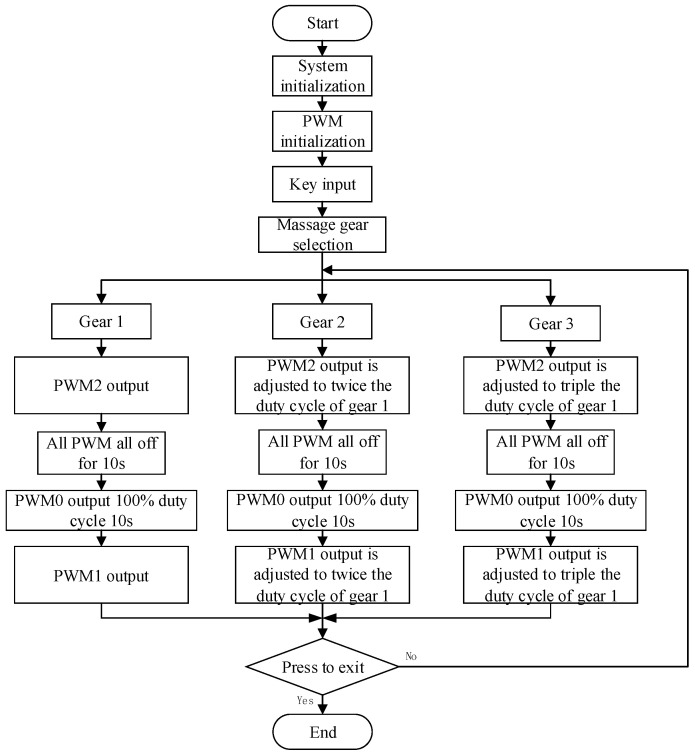
Working flow chart of the uni-acupoint massage device.

**Figure 6 micromachines-13-00370-f006:**
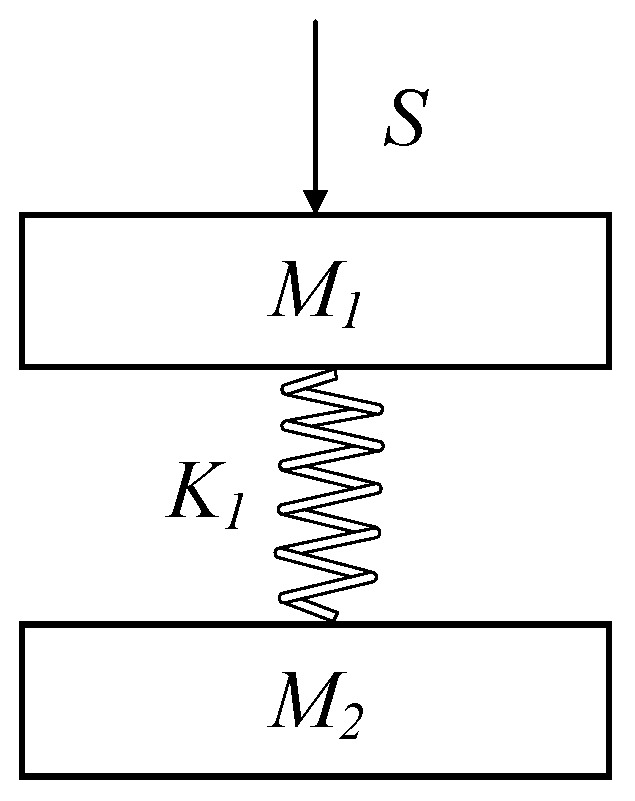
Simplified dynamic model of the uni-acupoint massage device.

**Figure 7 micromachines-13-00370-f007:**
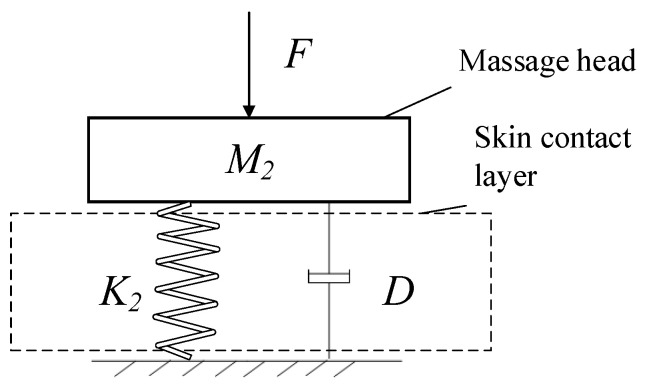
Equivalent model of human skin compression.

**Figure 8 micromachines-13-00370-f008:**
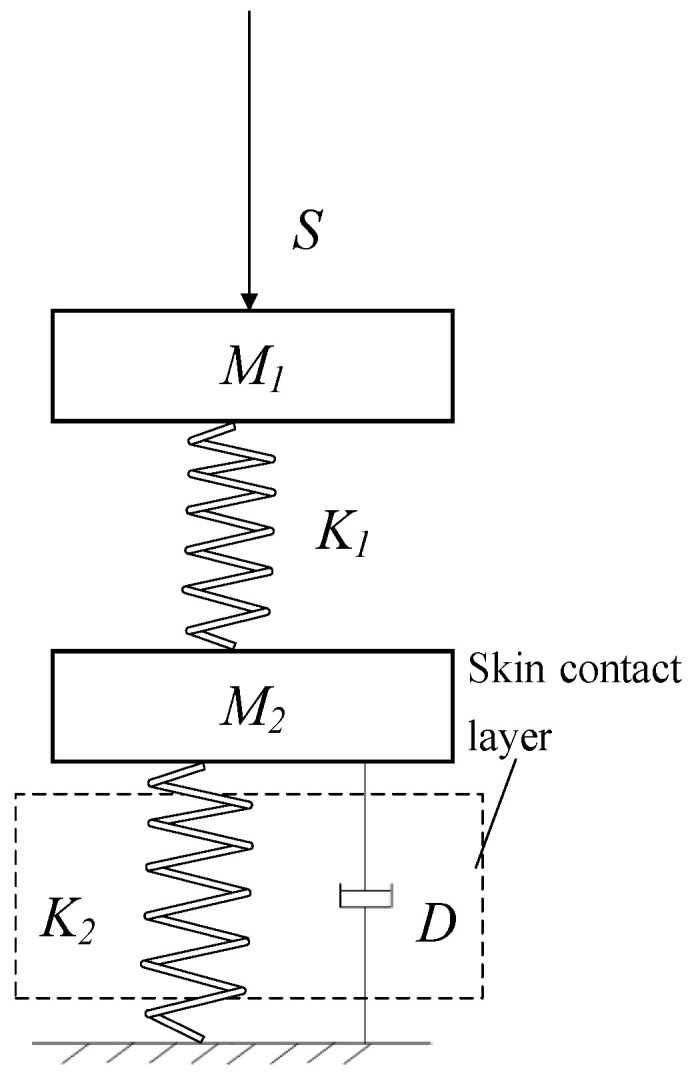
Series elastic actuating mechanism–human equivalent model.

**Figure 9 micromachines-13-00370-f009:**
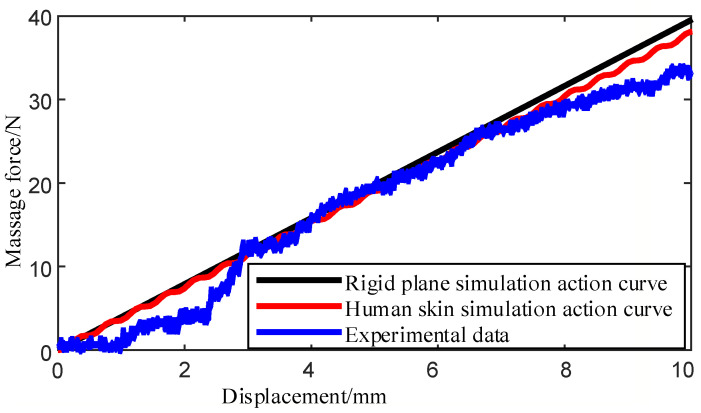
Fitting diagram of the prototype experimental displacement–force curve and simulation experimental curve.

**Figure 10 micromachines-13-00370-f010:**
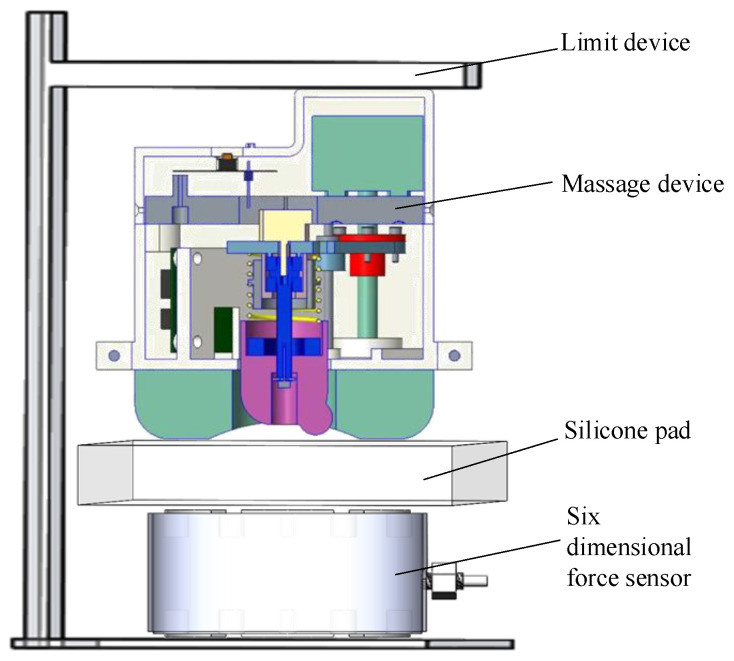
Massage force test of the uni-acupoint massage device.

**Figure 11 micromachines-13-00370-f011:**
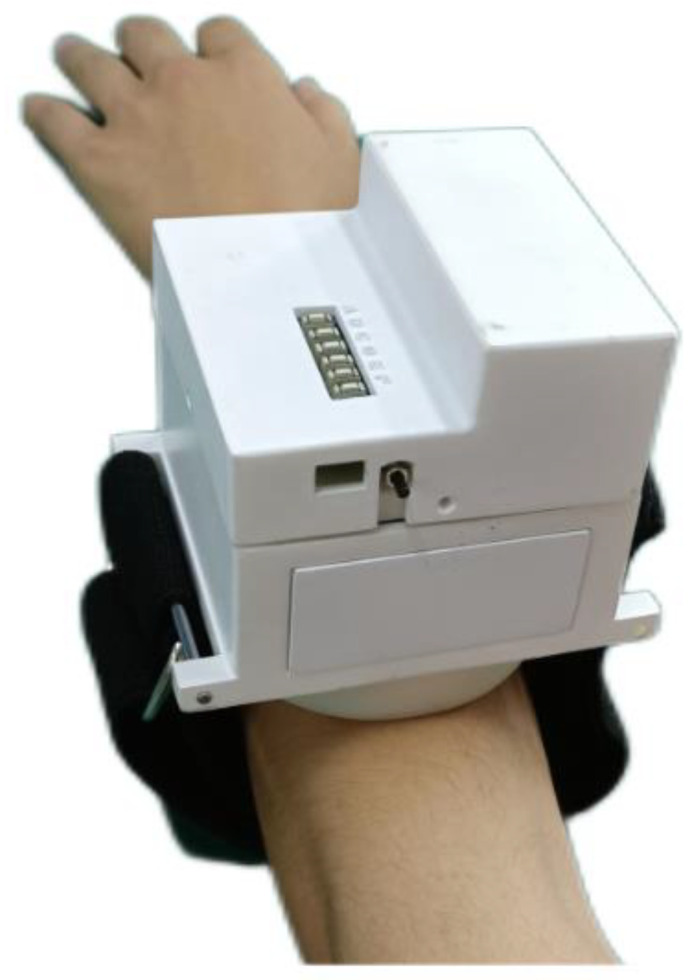
Wearing experiment for the uni-acupoint massage device.

**Table 1 micromachines-13-00370-t001:** Summary of massage manipulation characteristics.

Technique Name		Action Site	Direction	Freedom	Intensity	Frequency (Times/s)
pressing method	palm-pressing	palmar surface	vertical	1	20~80 N	0.17~0.25
	finger-pressing	finger surface	vertical	1	10~80 N	0.05~0.17
kneading method	palm- kneading	palmar surface	rotate	1	20~60 N	1~1.3
	finger- kneading	finger surface	rotate	1	20~80 N	1~1.3

**Table 2 micromachines-13-00370-t002:** Technical indexes of the acupoint massage device.

Number	Process	Index
1	finger-pressing	contact force 10–80 N holdup time 10 s
2	finger- kneading	contact force 10–80 N rotation speed 60 rpm rotation time 10 s
3	releasing	relax 10 s

## Data Availability

Not applicable.

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
