# Peer review of "Design and Massaging Force Analysis of Wearable Flexible Single Point Massager Imitating Traditional Chinese Medicine"

_micromachines, 2022, doi:10.3390/mi13030370_

Round 1
Reviewer 1 Report
This paper explores a novel problem: wearable flexible single point massager, which has a certain reference significance. The paper can be accepted after minor revision. Therefore, the comments should be first considered as follows:
1. [Line 91] The purpose of this is not to make the acupoints tired. What does this sentence mean? Please have someone competent in the English language and the subject matter of your paper go over the paper and correct it.
2.[Line 131] “In order to control the contact force of the massage device, … , but it will undoubtedly increase the cost. ”Please indicate whether the conclusion is based on other papers or reprocessed by evaluating multiple published papers.
3.[Line 335] As this is a wearable device applied to limbs, the design of wearing experiment needs to be further improved. For example, supplement the wearing experiment of legs or the comparative experiment between healthy and non-healthy people.
4.[Figure 1,2] Before submitting the revision, please ensure that your materials have been properly prepared and formatted, and some picture formats need to be modified. For example, the identification methods in Figure 1 and Figure 2 are not consistent.
5. [Figure 11] The background which is unrelated to the experiment should not appear on the picture. It is recommended to use black or white picture background.
6. The literature review is not enough. The authors should review more recently published articles to clearly reflect their novelties. The authors should be encouraged to use more recently published papers (2017-2022). Furthermore, the references should have the same type, for instance, the name of references. One reference is full name, but the others are abbreviations. Moreover, the format of conferences should be unanimous.
Conclusion:
Therefore, I recommend it can be published when the above issues are corrected.
Author Response
Point 1: [Line 91] The purpose of this is not to make the acupoints tired. What does this sentence mean? Please have someone competent in the English language and the subject matter of your paper go over the paper and correct it.
Response 1: Thank you for your constructive suggestions. The original meaning of this sentence is to summarize the previous description of massage techniques and put forward the significance of doing so, but our wording may not be accurate enough, which has caused your misunderstanding. The purpose of alternating different massage methods described above is to prevent acupoints from becoming insensitive or even easily damaged due to maintaining the same massage method for a long time. After carefully considering your comments, we changed the description of this sentence and checked the full text again to ensure that such errors do not occur again.
Point 2:[Line 131] “In order to control the contact force of the massage device, … , but it will undoubtedly increase the cost. ”Please indicate whether the conclusion is based on other papers or reprocessed by evaluating multiple published papers.
Response 2: Thank you for pointing this out in the manuscript. As commented by the reviewer, the control of friction force is studied and discussed in this paper, but there are only conclusive words, which is our negligence. This conclusion is based on the previous review of relevant research. The control of contact force generally requires the addition of expensive force sensors, which will increase the cost. Of course, there are also literatures describing this problem. For this, we add an article as a reference.
Point 3: [Line 335] As this is a wearable device applied to limbs, the design of wearing experiment needs to be further improved. For example, supplement the wearing experiment of legs or the comparative experiment between healthy and non-healthy people.
Response 3: The reviewer' opinion is very important. Indeed, we have completed the experiment of wearing limbs, in which the results of the experiment of wearing legs are similar to those of wearing arms. For reasons of representativeness, we only chose the experiment of wearing arms in the manuscript. The comparative experiment is indeed a very meaningful experiment, but it needs a long observation period, which will be the focus of our next research.
Point 4: [Figure 1,2] Before submitting the revision, please ensure that your materials have been properly prepared and formatted, and some picture formats need to be modified. For example, the identification methods in Figure 1 and Figure 2 are not consistent.
Response 4: We have corrected it and we also feel great thanks for your point out. First of all, we have fully prepared the material, and the format of the material has been repeatedly checked to ensure that the format is correct. Secondly, we have unified modified the picture format of the full text to ensure that the picture format of the full text is consistent, and we have modified the picture content appropriately to make the meaning of the picture clearer. The corrected picture is placed below for your convenience.
Figure 1. (a)
Figure 2
Figure 3
Point 5: [Figure 11] The background which is unrelated to the experiment should not appear on the picture. It is recommended to use black or white picture background.
Response 5: Thanks to reviewers for their comments. We have changed the experimental background to white to ensure that non-experimental backgrounds do not appear in the image.
Figure 11
Point 6: The literature review is not enough. The authors should review more recently published articles to clearly reflect their novelties. The authors should be encouraged to use more recently published papers (2017-2022). Furthermore, the references should have the same type, for instance, the name of references. One reference is full name, but the others are abbreviations. Moreover, the format of conferences should be unanimous.
Response 6: Thank you for your valuable comments. We regret to cause you trouble in the use of reference. We added some new references,and some papers published five years ago still have typical significance, which is the reason why we choose them. We have modified and checked the citation type of the paper to ensure that its citation format is consistent. We have also unified the conference form of the paper to make it consistent.

Reviewer 2 Report
[Line 55] The authors argue that large devices can not be accurately applied to specific areas such as limbs. There is a clear counter-argument of this: patients can move their limbs really while thus can very accurately place the limbs to large massaging machines. This needs to be logically addressed.
[Line 82] please specify whether table one is cited from other papers, experimentally measured by the authors, or reprocessed by assessing multiple published papers.
[Line 132] One of the clearest separators of hand pressing vs machine is speed and acceleration. In addition to the frequency and pressure, it is also important to explain and explore the choice of moving speed of the pressing motor. Although many motors can mimic the force and frequency of the human hand very accurately, it is much harder to measure the speed/acceleration of every single press, which machines often do much faster than human hands.
[Line 150] Please explain why the massage head has that specific shape and preiection: specifically, please explain how you decided the size and angle of the preiection.
[Figure 6, 7] Please check in with journal format for drawing: in some unified physical illustrations, spring is represented by round-cornered zig-zag lines; the presented illustration can be confused with a resister.
[Figure 11] according to the figure, it appears that the device is rather big. Please verify (with video or time-lapse) that the device does not have undesired movements when worn on limbs and turned on.
Author Response
Point 1: Line 55 The authors argue that large devices can not be accurately applied to specific areas such as limbs. There is a clear counter-argument of this: patients can move their limbs really while thus can very accurately place the limbs to large massaging machines. This needs to be logically addressed.
Response 1: Many thanks to reviewers for their valuable comments on our work. Compared with the device you mentioned, the significance of our device is that it is small and light enough for users to easily wear it. We have already mentioned many typical massage devices before. On the basis of the survey, we found that although there are patients who can move their limbs and place them accurately on a large massager, it is difficult for a large massager to continuously and efficiently massage a single acupoint. And according to some massage devices mentioned in the paper and things in two new added literature, it can be concluded that the current large-scale massage devices are mostly for back massage. But the massager proposed in this paper can continuously and efficiently massage individual points of human limbs. According to the comments of reviewers, we have revised the manuscript to ensure its logic is correct.
Point 2: Line 82 please specify whether table one is cited from other papers, experimentally measured by the authors, or reprocessed by assessing multiple published papers.
Response 2: Thank you for your comments. The data in Table 1 are derived from reference 16 and 17. We have revised the manuscript according to the suggestions of reviewers to make it clearer.
Point 3: [Line 132] One of the clearest separators of hand pressing vs machine is speed and acceleration. In addition to the frequency and pressure, it is also important to explain and explore the choice of moving speed of the pressing motor. Although many motors can mimic the force and frequency of the human hand very accurately, it is much harder to measure the speed/acceleration of every single press, which machines often do much faster than human hands.
Response 3: Thank you very much for your valuable comments on our work. This is a very important question. Obviously, under different speeds and accelerations, the massage effect of the massager will be different. However, there is no study on different speeds and accelerations in this paper. In this paper, we just investigated the rotation speed of the hand of the masseur in the massage process, simulated it in the design, and set the speed to a constant 66 rpm. In the next step, we will focus on the impact of different speed and acceleration of the massager on the massage effect of the massager.
Point 4: Line 150 Please explain why the massage head has that specific shape and preiection: specifically, please explain how you decided the size and angle of the preiection.
Response 4: Thank you for your constructive comments on our work. In traditional Chinese medicine finger kneading massage, the point at which the masseur applies force is not fixed, but rotates around the fixed axis, and the massage force they apply to the patient through the finger abdomen (to increase the force area) is not perpendicular to the contact surface, but has an inclination angle of nearly 45 degrees with the contact surface. Therefore, the designed massage head needs to have a bias convex massage point. The massage point can be divided into two parts. One is a semicircle for connecting with the massage head, and the other one is an ellipse with large curvature for imitating the finger abdomen. At the same time, there is a large included angle between the massage point and the middle shaft of the spring within the allowable range of the mechanism.
Point 5: Figure 6, 7 Please check in with journal format for drawing: in some unified physical illustrations, spring is represented by round-cornered zig-zag lines; the presented illustration can be confused with a resister.
Response 5: The reviewer's opinion is very important. According to the suggestions of reviewers, we modified the expressions of all springs in the manuscript to be represented by rounded corners and serrated lines. And unified the format of all the pictures of the manuscript.
Figure 6
Figure 7
Figure 8
Point 6: Figure 11 according to the figure, it appears that the device is rather big. Please verify (with video or time-lapse) that the device does not have undesired movements when worn on limbs and turned on.
Response 6: Thank you very much for your valuable advice. First of all, please allow us to apologize for this point. Due to the Chinese Spring Festival, we temporarily can't give you the recent record to meet your needs, we can only give you a video recorded in March 2021. It is an initial prototype work video, prototype and the equipment described in the manuscript has a certain difference, but the equipment appearance size and operation key components transmission relationship are consistent. We hope this video can solve your problem. In this video, you can find that the device is similar in size to the arm of most adults, slightly smaller than the leg, and the massaging head is similar in size to the thumb. In general, the device is not large enough and the size meets the requirements described in the manuscript.
